# Four-top quark physics at the LHC

**Freya Blekman** [1,*] , **Fréderic Déliot** [2] , **Valentina Dutta** [3] **and Emanuele Usai** [4]

1     Deutsches Elektronen-Synchrotron DESY, Notkestr. 85, 22607 Hamburg, Germany, and Universität Hamburg, Luruper Chaussee 149, 22761 Hamburg, Germany
2     Irfu, CEA Paris-Saclay, Université Paris-Saclay, CEDEX, 91191 Gif-sur-Yvette, France
3     Physics Department, Carnegie Mellon University, Wean Hall, 5000 Forbes Avenue, Pittsburgh, PA 15213-3890, USA
4     Department of Physics and Astronomy, The University of Alabama, 514 University Blvd, Tuscaloosa, AL 35487-0324, USA
*     Correspondence: freya.blekman@desy.de

**Abstract:** The production of four top quarks presents a rare process in the Standard Model that provides unique opportunities and sensitivity to Standard Model observables including potential enhancement of many popular new physics extensions. This article summarises the latest experimental measurements of the four-top quark production cross section at the LHC. An overview is provided detailing interpretations of the experimental results regarding the top quark Yukawa coupling in addition to the limits on physics beyond the Standard Model. Further, prospects for future measurements and opportunities offered by this challenging final state are given herein.

**Keywords:** four top quarks; top quark; top quark Yukawa coupling; LHC; ATLAS; CMS; particle physics; SMEFT; collider physics; hep-ex; quantum chromodynamics





## 1. Introduction

The top quark was discovered in 1995 [1,2] and plays a pivotal role in particle physics at the energy frontier. In the LHC era, top quark pair production is under high scrutiny by the ATLAS and CMS collaborations [3–7]. Single-top quark production, originally observed at the Tevatron, is now also well established at the LHC [8–14]. Although studies surrounding the intrinsic physics properties represent an important area of research, top quarks play a key background role in many analyses that concern the search for new physics signatures.

The simultaneous production of four top quarks is an example of a rare multiparticle process in the Standard Model (SM), and also presents a promising avenue in the search for signals of new physics beyond the Standard Model (BSM). The production of four top quarks is interesting in its own right since experimental data are expected to challenge state-of-the-art perturbative QCD calculation techniques. A selection of representative diagrams is presented in Figure 1. Recent advanced calculations predict the $t\bar{t}t\bar{t}$ cross section at a centre-of-mass energy of $\sqrt{s} = 13$ TeV to be 12.0 $\pm$ 2.4 fb at next-to-leading order in QCD including NLO electroweak corrections, with the quoted uncertainty originating from renormalisation and factorisation scales [15–20]. When focusing on events with two $t\bar{t}$ pairs, warranted questions arise concerning the relevance of double parton scattering in the search for $t\bar{t}t\bar{t}$ events. With a simple PYTHIA model at leading order [21], the cross section for this process can be confirmed to be of the order of 3 ab. As the cross section for $t\bar{t} + t\bar{t}$ double parton scattering is over three-orders of magnitude smaller than the $t\bar{t}t\bar{t}$ cross section, this background is thus irrelevant in the search for four-top quark production.

The $t\bar{t}t\bar{t}$ state provides direct ways to constrain otherwise tricky to measure SM parameters such as the top quark Yukawa coupling and several SM Effective Field Theory parameters sensitive to the quartic couplings between top quarks. If the scale of new physics is beyond the capacity of direct observation, it can manifest as a deviation from the

SM, e.g., a modification of the $t\bar{t}t\bar{t}$ cross section created by virtual and direct (s-channel) contributions of undiscovered BSM particles. These measurements would provide crucial input to the understanding of the SM.

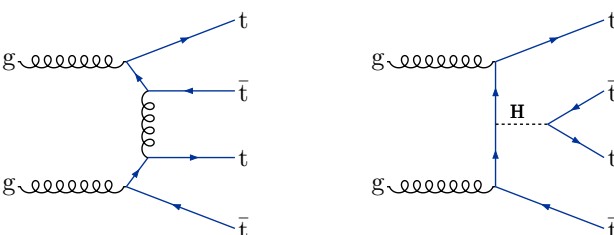

**Figure 1.** Selected Feynman graphs representing the main production modes of $t\bar{t}t\bar{t}$ production.

The current state-of-the art search approaches for $t\bar{t}t\bar{t}$ production show that the LHC Run 2 dataset is sufficient to establish evidence [22]. Due to the presence of four top quarks in the event, and consequently four W bosons and four b-quarks, the experimental challenges of the search for $t\bar{t}t\bar{t}$ production depend heavily on the W boson decay considered. Analyses considering multi-lepton and same-charge dilepton decays typically have the highest impact in the search significance and are characterised by a very low-branching fraction and acceptance, which is compensated by extremely low backgrounds from SM particles. On the other hand, the analyses of single-lepton and opposite-charge dilepton bear much higher branching fractions but considerable background from top quark pair production. Finally, the all-hadronic final state is largely driven by the reduction in overwhelming background from QCD multijet production.

This review paper summarises the current experimental status in the study of $t\bar{t}t\bar{t}$ production. It is also essential to look forward as $t\bar{t}t\bar{t}$, when firmly established, has substantial physics potential for future HL-LHC runs [23,24] and future hadron colliders [25]. Section 2 summarises the current status of $t\bar{t}t\bar{t}$ measurements. In Sections 3 and 4, interpretations of the SM measurements, BSM searches and opportunities for further analyses are discussed.

## 2. Current Status of Four-Top Quark Measurements

Searches for $t\bar{t}t\bar{t}$ production in proton–proton (pp) collisions at a center-of-mass energy of 13 TeV were conducted by the ATLAS and CMS collaborations in multiple final states. The most recent results, which supersede previous ones, are described below. The most sensitive searches from ATLAS and CMS target same-charge dilepton and multi-lepton final states [22,26] (Section 2.1), and were carried out with data collected during Run 2 of the LHC. The data samples used for these searches correspond to 139 fb$^{-1}$ for ATLAS, and 137 fb$^{-1}$ for CMS. Searches targeting single-lepton and opposite-charge dilepton final states [27,28] have also been conducted by both collaborations (Section 2.2). Finally, a search in the all-hadronic final state has also recently been conducted by CMS (Section 2.3).

### 2.1. Searches for $t\bar{t}t\bar{t}$ Production in Same-Charge Dilepton and Multi-Lepton Final States

These searches target final states with either two same-charge or at least three light-charged leptons (electrons or muons), corresponding to a combined branching fraction of $\approx$12% for $t\bar{t}t\bar{t}$ events. Since these final states have low levels of background from other SM processes, they are the most sensitive to $t\bar{t}t\bar{t}$ production.

#### 2.1.1. Event Selection and Backgrounds

In addition to the lepton requirements, events selected for the searches are required to have jet activity consistent with the hadronization of b quarks from the top quark decays, or from hadronic decays of the W bosons that do not decay leptonically, as well as large overall event activity. A minimum requirement of at least two jets ($N_{\mathrm{jet}} \geq 2$) is imposed for the CMS search, while a more stringent requirement of $N_{\mathrm{jet}} \geq 6$ is imposed for the ATLAS search. The difference in $N_{\mathrm{jet}}$ requirements between the two experiments is driven by

whether control regions are defined separately or included directly in the baseline selection. Both searches require at least two jets to be "tagged", or identified, as b-jets ($N_b \geq 2$). For the ATLAS search, a requirement of large event activity is imposed by requiring the scalar sum of the transverse momenta of jets and isolated leptons to exceed 500 GeV, while for the CMS search, a minimum requirement of 300 GeV is imposed on the the scalar sum of the transverse momenta of jets. The CMS search also requires the presence of missing transverse momentum ($p_T^{\text{miss}} > 50$ GeV). The latter is expected to arise from the presence of neutrinos from leptonic W boson decays, which would escape the detector without leaving a visible signature.

Backgrounds to these searches arise from processes in which $t\bar{t}$ is produced in association with bosons that decay leptonically, i.e., $t\bar{t}W$, $t\bar{t}Z$, and $t\bar{t}H$ production, particularly when these processes are accompanied by the production of additional jets. These backgrounds are generally estimated using simulated events. In ATLAS, the $t\bar{t}W$ background receives a different treatment because theoretical studies [15,19,29–35] showed that electroweak corrections not included in the used simulation have a significant effect. Previous measurements [36] also showed that $t\bar{t}W$ production in association with jets could obtain a larger normalisation factor than predicted by the Monte Carlo (MC) simulation. For these reasons, normalisation of the $t\bar{t}W$ background in the ATLAS analysis is corrected using data in a dedicated control region (CR). In CMS, a dedicated CR is used to constrain normalisation of the $t\bar{t}Z$ background. The simulated samples are corrected to account for observed discrepancies in CMS data. In particular, the modeling of the multiplicity of additional jets from initial- or final-state radiation (ISR or FSR) in $t\bar{t}Z$ and $t\bar{t}W$ simulation is improved by reweighting the ISR/FSR jet multiplicity. Additionally, the modeling of the flavour of additional jets in $t\bar{t}W$, $t\bar{t}Z$, and $t\bar{t}H$ simulation is corrected based on the measured ratio of $t\bar{t}b\bar{b}$ and $t\bar{t}jj$ events, $1.7 \pm 0.6$ [37], where $j$ represents a jet of any flavour.

Backgrounds may also originate from dilepton $t\bar{t}$ decays with one lepton that has an erroneously assigned charge, or from single-lepton $t\bar{t}$ decays with an additional "non-prompt" lepton. Here, a non-prompt lepton refers to a lepton produced in a hadron decay or from a photon conversion in a jet, or to a hadronic jet that is misidentified as a lepton. The background with charge-misidentified electrons is estimated by applying the electron charge-misidentification probabilities measured in simulation and corrected to account for discrepancies with data (or directly measured in data using $Z \rightarrow ee$ events) to opposite-charge dilepton events. The charge-misidentification probability for muons is an order of magnitude smaller, and therefore the background with charge-misidentified muons is considered to be negligible.

For the ATLAS measurement, the non-prompt lepton background is estimated using the so-called template method. This method relies on the simulation to model the kinematic distributions of background processes arising from non-prompt leptons and on CRs to determine their normalisations. These CRs are included in the fit together with the signal region (SR), and the normalisation factors are determined simultaneously with the $t\bar{t}t\bar{t}$ signal. For the CMS search, the non-prompt lepton background is estimated using the "tight-to-loose" ratio method [38]. The more stringent ("tight") lepton selection criteria used in the SRs are relaxed to define a "loose" selection enriched in non-prompt leptons. The efficiency of non-prompt leptons satisfying the "loose" criteria to also satisfy the "tight" criteria is measured in a control sample. The non-prompt lepton background contribution in the SRs is then estimated by applying weighting factors to events selected by requiring at least one lepton to pass the loose selection while failing the tight one.

### 2.1.2. Signal Extraction and Results

The ATLAS search separates signal from background events using a multivariate discriminant built in the signal region. The most important inputs to the boosted decision tree (BDT) are the best pseudo-continuous b-tagging discriminant scores [39] summed over all the jets in the event as well as the minimum distance between two leptons among all possible pairs. The $t\bar{t}t\bar{t}$ production cross section and the normalisation factors of the

backgrounds are determined via a binned likelihood fit to the BDT score distribution in the SR and to discriminating variable distributions in background CRs. The systematic uncertainties in both the signal and background predictions are included as nuisance parameters. The measured $t\bar{t}t\bar{t}$ production cross section is $\sigma(t\bar{t}t\bar{t}) = 24 \pm 5\text{(stat)}^{+5}_{-4}\text{(syst)}$ fb $= 24^{+7}_{-6}$ fb. The significance of the observed (expected) signal is found to be 4.3 (2.4) standard deviations. The normalisation factors of the different background sources determined from the fit are compatible with 1 except for $t\bar{t}W$. Apart from the theoretical uncertainty of the signal cross section, the largest systematic uncertainty impacting the signal extraction originates from the modelling of the $t\bar{t}W$ + jets process. Within the uncertainties of the background modelling, the impact of the uncertainty in $t\bar{t}t\bar{t}$ production is also significant. The distribution of the BDT score in the SR after performing the fit is shown in Figure 2 (left) where a good agreement between data and the fitted prediction is observed.

In the CMS search, a BDT classifier is trained to distinguish $t\bar{t}t\bar{t}$ from background events, using variables that include $N_\text{jet}$, $N_\text{b}$, $N_\text{l}$, $p_\text{T}^\text{miss}$, $H_\text{T}$ (scalar sum of jet transverse momenta) and other kinematic properties of the jets and leptons in an event. Events are subdivided into 17 SRs based on the BDT discriminant output. Based on the results of a binned maximum-likelihood fit to the data combining all exclusive SRs and the $t\bar{t}Z$ CR, in which nuisance parameters representing systematic uncertainties are profiled, the cross-section measurement for $t\bar{t}t\bar{t}$ production is $\sigma(t\bar{t}t\bar{t}) = 12.6^{+5.8}_{-5.2}$ fb. The observed (expected) significance relative to the background-only hypothesis is 2.6 (2.7) standard deviations. Figure 2 (right) shows the distribution of events in the SRs and CR included in the fit for the BDT analysis, with the post-fit estimates for background and signal.

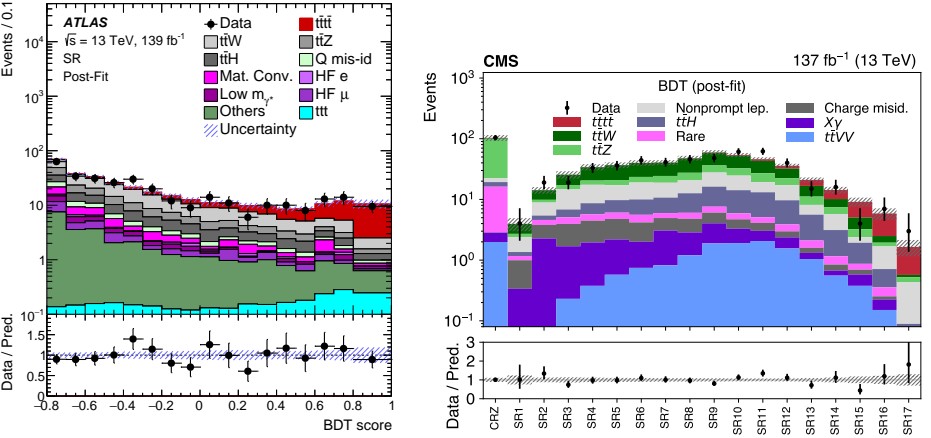

**Figure 2.** Comparison between data and prediction after the fit for the distribution of the BDT score in the signal region of the ATLAS multi-lepton analysis [22] (**left**), and for events in the $t\bar{t}Z$ CR and SRs of the CMS BDT-based multi-lepton analysis (**right**) [26].

### 2.2. Searches for $t\bar{t}t\bar{t}$ Production in Single-Lepton and Opposite-Charge Dilepton Final States

The target events of this search contain either exactly one (1L) or exactly two opposite-charge light-charged leptons (2LOS). In the latter case, the leptons can have different flavour. The total branching fraction of these final states is about 57% of the $t\bar{t}t\bar{t}$ events. Despite the much larger branching fraction with respect to the analysis in Section 2.1, this combination of final states nears lower sensitivity. This is caused by large cross-section SM processes with similar final states such as $t\bar{t}$ production with additional jets. In particular, $t\bar{t}$ + bb production is the major background and the proper modelling of this process and its separation from $t\bar{t}t\bar{t}$ are major challenges of this analysis.

#### 2.2.1. Event Selection and Backgrounds

The final state is characterised by four b-quarks resulting from the decays of the four top quarks and by either six or four light jets arising from the hadronic decays of the W

boson decays and also from the top quark decays. Thus, in ATLAS, events are required to have at least 10 jets (8 jets) in the 1L channel (2LOS channel), among which four are b-tagged. In CMS, events are required to have at least seven jets (muon channel) and eight jets (electron channel) for the 1L final state and at least four jets for the 2LOS final state. The difference in $N_{jet}$ requirements between the two experiments is once again driven by whether control regions are defined separately or included in the simultaneous fit. The background in the high jet multiplicity regions was found to be mis-modelled by MC, thus ATLAS developed a strategy to reweight the $t\bar{t}$ MC generation using data to obtain a reliable $t\bar{t}$ +jets estimate. In addition, the rate of $t\bar{t}$ production in association with $b$-jets was observed to be underestimated in the MC simulation, so it is adjusted as well. The selected events are categorised according to the lepton and jet multiplicities with different b-tagging requirements. Corrections to the normalisation and kinematics of the $t\bar{t}$ + light, $t\bar{t} + \geq 1c$ and $t\bar{t} + \geq 1b$ jets are derived using data in regions with 2 $b$-tagged jets where there is low signal contamination and validated in regions with 3 $b$-tagged jets. The first reweighting adjusts the normalisation of $t\bar{t}$ production with heavy flavour jets. A sequential reweighting is then performed to mitigate the kinematic mismodelling in the distributions of number of jets, number of jets with large radius, scalar sum of all jet and lepton momenta and the average angular separation between two jets. In CMS, a dedicated correction for the modeling of $t\bar{t}$ in high jet multiplicity events is derived in a signal depleted region and applied to signal enriched regions. Additionally, the top quark transverse momentum spectrum of $t\bar{t}$ simulated events is corrected to match the observed spectrum in data.

### 2.2.2. Signal Extraction and Results

In the ATLAS analysis, the different $t\bar{t}$ +jets components after reweighting are further adjusted and constrained in a binned profile likelihood fit together with the extraction of the signal strength. A total of 21 control and signal regions are used in the fit (12 regions in the 1LOS channel and 9 regions in the 2LOS region). In the region most sensitive to $t\bar{t}t\bar{t}$ production, BDTs are used to discriminate signal from background events after applying reweighting. Several variables are inputs to the discriminant: global event variables, and kinematic properties of the reconstructed objects. Among the input variables, jets with large radius are used as proxies for hadronically decaying top quark with high momentum. The most powerful variable across all regions is the sum of the b-tagging score of the six jets with the highest scores. Several uncertainties are implemented as nuisance parameters in the fit and special care is taken for uncertainties in the $t\bar{t}$ background prediction since these uncertainties have the largest impact on the measurement sensitivity. Following the fit, the $t\bar{t}t\bar{t}$ cross section is measured to be: $\sigma(t\bar{t}t\bar{t}) = 26 \pm 8(\text{stat})^{+15}_{-13}(\text{syst})$ fb $= 26^{+17}_{-15}$ fb which corresponds to an observed significance of 1.9 standard deviations relative to the background-only hypothesis (while 1.0 standard deviation is expected). The largest systematic uncertainty is revealed to originate from the modelling of $t\bar{t} + \geq 1b$ jets, mainly driven by the generator and flavour scheme uncertainty. The observed and expected event yields are shown in Figure 3.

This measurement is further combined with the result in the same-charge dilepton and multi-lepton channel (see Section 2.1) by performing a simultaneous profile likelihood fit across all regions of both analyses. Most of the relevant systematic uncertainties in these two analyses are uncorrelated. The combined $t\bar{t}t\bar{t}$ cross section is measured to be $\sigma(t\bar{t}t\bar{t}) = 24 \pm 4(\text{stat})^{+5}_{-4}(\text{syst})$ fb $= 24^{+7}_{-6}$ fb. The observed (expected) significance of the result is $4.7\sigma$ ($2.6\sigma$) above the background-only hypothesis, presenting an improvement over the result in the same-charge dilepton and multi-lepton channel alone.

In CMS, events are categorised as a function of their jet multiplicity, b-tagged jet multiplicity, and top-tagged jet multiplicity (using a BDT algorithm to identify hadronically decaying top quarks). Different multiplicity ranges are considered for the single-lepton and the dilepton final states.

In order to reduce background from QCD multijet processes, in addition to the event categorisation, CMS requires that $H_T > 500$ GeV and $p_T^{miss} > 50$ GeV are imposed. The

single-lepton analysis uses event-level BDTs to discriminate $t\bar{t}t\bar{t}$ events from the predominant $t\bar{t}$ background. The event-level BDT is trained using global event variables and employs top and bottom tagger outputs in addition to jet kinematics and angular relationships between leptons and jets. The dilepton final states analysis uses $H_T$, the sum of the transverse momentum of all jets in the event, except for the analysis performed on the 2016 data which follows the same strategy as the single-lepton analysis and uses a BDT. A binned likelihood fit to the event-level distributions is used to set limits and best fit to the $t\bar{t}t\bar{t}$ cross section and determine the significance of the signal over the no $t\bar{t}t\bar{t}$ hypothesis. The single-lepton state analysis has an observed significance of 1.2 standard deviations, while the expected significance from simulation, assuming the SM $t\bar{t}t\bar{t}$ cross section, is 1.4 standard deviations. The measured best fit to the signal cross section is $15^{+13}_{-11}$ fb. The opposite-sign dilepton state analysis has an observed significance of 1.8 standard deviations, while the expected significance from simulation, assuming the SM $t\bar{t}t\bar{t}$ cross section, is 0.6 standard deviations. The measured best fit to the signal cross section is $37^{+21}_{-20}$ fb. These are combined with all other final states which brings a significant improvement in the cross section limits and best fit measurement, described in Section 2.3.3.

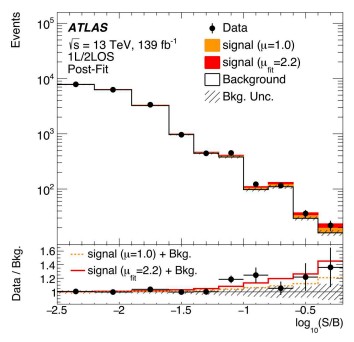 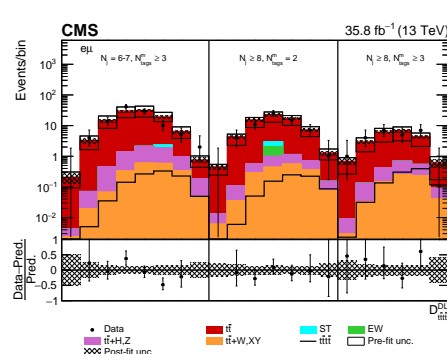

**Figure 3.** Observed and expected event yields as a function of $\log_{10}(S/B)$ where S and B are the post-fit signal and background yields in the single-lepton and opposite-charge dilepton ATLAS analysis [27] (**left**). Post-fit and observed distribution of the event-level BDT in three separate signal-enriched regions in the $e\mu$ final state, from the single-lepton and opposite-charge dilepton CMS analysis on the 2016 data (**right**) [28].

### 2.3. Search for $t\bar{t}t\bar{t}$ Production in the All-Hadronic Final State

For $t\bar{t}t\bar{t}$ measurements, the final state in which all four W bosons from the top quark decays subsequently decay hadronically represents a challenging but important exploration opportunity. Roughly 20% of $t\bar{t}t\bar{t}$ events are expected to decay into the all-hadronic final state. The main challenge of this final state lies in the experimental backgrounds: a very large background arises from purely QCD multijet events, while another significant source of background originates from $t\bar{t}$ events with fully hadronic top quark decays and additional jets. The QCD multijet background is especially challenging to model accurately with simulation, and thus data-driven methods are needed in order to obtain a robust background prediction in this final state. This final state for $t\bar{t}t\bar{t}$ measurements was recently studied for the first time by CMS [40].

#### 2.3.1. Event Selection and Backgrounds

Events selected for this search are required to have no identified leptons, a minimum of nine jets with at least three of them being b-tagged, and $H_T > 700$ GeV. In order to distinguish potential signal events from the multijet background, the search makes use of dedicated techniques to identify the presence of hadronically decaying top quarks with either moderate or large Lorentz boosts. Moderately boosted hadronic top quark decays will typically produce three separate jets in the detector; these "resolved" top quark decays are identified with a custom BDT-based algorithm. In contrast, the decay products of

significantly boosted top quarks can be reconstructed in a single large-radius jet, and are identified with the algorithm defined in Ref. [41]. In order to avoid double counting, resolved and boosted top quark candidates are required to be well separated in $\eta - \phi$ space. A minimum requirement of at least one resolved top candidate ($N_{RT} \geq 1$) is imposed in selecting events for this search. Events are then categorized into 12 exclusive SR categories based on $H_T$, $N_{RT}$, and the number of boosted top quark candidates ($N_{BT}$).

The dominant backgrounds in the search originate from hadronic $t\bar{t}$ decays and QCD multijet events. Data-driven methods are employed to estimate the normalization and BDT shape for these backgrounds. An extension of the ABCD method [42] is applied to estimate the number of background events in the SR categories. A total of five CRs, defined using $N_{jet}$ and $N_b$, are used for the estimation—two more than in the traditional ABCD method, in order to better account for correlations between the variables and higher-order effects. The background BDT shape in the SRs is predicted using a Deep Neural Network (DNN) [43]. The DNN is trained in the five CRs to learn shape transformations from $t\bar{t}$ simulation to the estimated QCD multijet plus $t\bar{t}$ shape in data (after the subtraction of other background contributions estimated from simulation). The $H_T$ and BDT shapes are learned simultaneously in each $N_{RT}$ and $N_{BT}$ category, and the DNN is then used to predict the combined shape of the QCD multijet and $t\bar{t}$ background in the SR. Small additional background contributions originate from $t\bar{t}W$, $t\bar{t}Z$, and $t\bar{t}H$ events and from diboson production. These are estimated from simulation.

### 2.3.2. Signal Extraction and Results

An event-level BDT is used to extract signal from background in the SR via a simultaneous maximum-likelihood fit to the BDT shape in all SR categories. The BDT input variables include the multiplicity and kinematics of jets and b-tagged jets, the kinematics of top-tagged candidates, variables related to jet angular distributions, and event shape variables. The expected significance for this analysis is $0.4\sigma$; however, a non-significant excess of $2.5\sigma$ above background is observed. This corresponds to a $t\bar{t}t\bar{t}$ production cross section of $\sigma(t\bar{t}t\bar{t}) = 70^{+30}_{-29}$ fb.

### 2.3.3. CMS Run 2 Combination

For LHC Run 2, the all-hadronic, single-lepton, opposite and same-charge dilepton and multi-lepton channel $t\bar{t}t\bar{t}$ measurements by the CMS Collaboration were explicitly designed to select orthogonal kinematic phase space. This also meant that control regions were chosen so as to not partially overlap with signal regions in other final states. This strategy allows all final states except those explicitly containing $\tau$ leptons to be combined by performing a simultaneous profile likelihood fit. The systematic uncertainties were correlated when appropriate. The combined $t\bar{t}t\bar{t}$ cross section is measured to be $\sigma(t\bar{t}t\bar{t}) = 17 \pm 5$ (stat+syst) fb, and the combination has an observed (expected) significance of $3.9\sigma$ ($3.2\sigma$) above the background-only hypothesis [40].

Despite the lower sensitivity of the all-hadronic, single-lepton and opposite-charge dilepton analyses compared to the same-charge dilepton and multi-lepton analysis, the combination of the different event signatures produces a significant improvement on the cross-section limits and best fit measurement. The systematic and statistical uncertainties are of similar magnitude, suggesting that in order to make further advances, improvements in analysis techniques are needed to suppress systematic uncertainties.

## 3. Interpretations

### 3.1. Yukawa Coupling

Four top-quark events can be produced with a virtual Higgs boson as mediator. So the $t\bar{t}t\bar{t}$ production rate is sensitive to the value of the coupling between the top quark and the Higgs boson ($y_t$) [44,45]. The advantages of the $t\bar{t}t\bar{t}$ process lie in the fact that it does not rely on any assumption on the Higgs width and that its cross section is proportional to the fourth-power of the top Yukawa coupling. It can also be used to probe the CP nature of

$y_t$. In addition to the $t\bar{t}H$ and $tH$ processes, $t\bar{t}t\bar{t}$ production can then help to shed valuable light on the Higgs boson properties.

CMS has used its upper limit on the measured $t\bar{t}t\bar{t}$ production rate from the multilepton channel described in Section 2.1 to constrain $y_t$. As the $t\bar{t}H$ background cross section also depends on $y_t$, the fit performed to extract the $t\bar{t}t\bar{t}$ cross section is repeated with the $t\bar{t}H$ contribution scaled by $|y_t/y_{SM}|$ where $y_{SM}$ is the SM value for the top quark Yukawa coupling. The resulting dependence on $y_t$ from the measured signal and background is then compared to the theoretical prediction obtained at LO [44] scaled to the NLO value of $12^{+2.2}_{-2.5}$ fb. The obtained 95% CL limits with the central, upper and lower values of the theoretical prediction are found to be $|y_t/y_{SM}| < 1.7, 1.4$ and $2.0$, respectively, [26]. Compared to the $|y_t/y_{SM}| < 1.6$ measured in differential $t\bar{t}$ production, these numbers are complementary and considered relatively model-independent compared to the values of $0.7 < |y_t/y_{SM}| < 1.1$ derived from direct measurements of $t\bar{t}H$ production [46–48]. Further analyses might investigate the use of $t\bar{t}t\bar{t}$ kinematics to better constrain $y_t$.

*3.2. EFTs*

Four top-quark production is sensitive to interactions between four heavy quarks (four-heavy-quark operators, QQQQ), to interactions between top quarks and light quarks (two-heavy-two-light four-quark operators, QQqq) and to operators that modify gluon–top quark interaction such as the chromomagnetic operator $c_{tG}$ (see for instance [49]). Among these, the four-heavy-quark operators can only be constrained by $t\bar{t}t\bar{t}$ or $t\bar{t}b\bar{b}$ production, which renders EFT studies in four top-quark production especially interesting. The QQqq operators affect $t\bar{t}t\bar{t}$ and $t\bar{t}$ production and consequently would also modify the backgrounds in $t\bar{t}t\bar{t}$ analyses (mainly the fake background and $t\bar{t}$ production in the same-charge dilepton and multi-lepton channel, or the $t\bar{t}$ +jets and $t\bar{t}b\bar{b}$ background in the single-lepton and opposite-charge dilepton channel).

There are five QQQQ operators that preserve $SU(2)_L$: $\mathcal{O}^1_{QQ}$, $\mathcal{O}^8_{QQ}$, $\mathcal{O}^1_{Qt}$, $\mathcal{O}^8_{Qt}$ and $\mathcal{O}^1_{tt}$. However, if we consider only the $t\bar{t}t\bar{t}$ process and LO operators, then these operators are redundant. Only four operators are independent and we can write: $\mathcal{O}^8_{QQ} = \frac{1}{3}\mathcal{O}^1_{QQ}$ (see for instance [45,50]).

From an experimental perspective, few analyses have interpreted the search for the four top quark process in the context of EFTs. An ATLAS search for four top quark production in the single-lepton and opposite-charge dilepton final state using a partial 13 TeV data set [51] performs such an interpretation. The EFT signal is modeled through a four-top quark contact interaction operator [52]. The normalization of the non-resonant signal is regulated by the expression $|C_{4t}|/\Lambda^2$ where $C_{4t}$ is the coupling constant and $\Lambda$ is the energy scale of new physics. The analysis set limits at 95% CL on $|C_{4t}|/\Lambda^2 < 1.9$ TeV$^{-2}$ (observed and expected).

The CMS search for four top quarks described in Section 2.1 reports an interpretation of the search for four top quark in terms of the Higgs oblique parameter $\hat{H}$. Within the context of an EFT, $\hat{H}$ is the Wilson coefficient of the only dimension-6 operator that modifies the Higgs boson propagator. This parameter modifies the off-shell behaviour of the Higgs boson. It can be proven that $t\bar{t}t\bar{t}$ is sensitive to $\hat{H}$ through the production modes containing the Higgs boson [53]. The CMS analysis uses simulations of the $t\bar{t}t\bar{t}$ process with modified $\hat{H}$ parameter. Additionally, the $t\bar{t}H$ cross section is scaled by a factor $(1 - \hat{H})^2$ to take into account the dependence on the oblique parameter. A 95% CL upper limit of $\hat{H} < 0.12$ is extracted from the analysis. The value is competitive with the constraint of $\hat{H} < 0.16$ [53] extracted using on-shell Higgs boson measurements [54].

The single-lepton plus opposite-charge dilepton analysis by CMS described in Section 2.2 studies the impact on $t\bar{t}t\bar{t}$ in EFT operators. Limits on the EFT operators are obtained neglecting any acceptance or BDT distribution shape deviations from the purely SM. At leading order, the four-top quark cross section in an EFT scenario can be parametrized as the SM cross section plus a combination of the coupling parameters of the four independent EFT operators that contribute to $t\bar{t}t\bar{t}$ $C_k$, where $k = \mathcal{O}^1_{tt}, \mathcal{O}^1_{QQ}, \mathcal{O}^1_{Qt}, \mathcal{O}^8_{Qt}$ with a set of four

linear parametrization coefficients $\sigma_k^{(1)}$, and a set of nine bilinear coefficients $\sigma_{k,j}^{(2)}$. The linear and bilinear parametrization coefficient are extracted from MADGRAPH simulations [17]. The parametrized formula of the cross section is used to extract expected and observed 95% CL intervals on the coupling coefficient of the four EFT operators. Limits are provided for two different scenarios, the "independent" scenario where only one coefficient is non-null and the "marginalized" scenario where the other coefficients are constrained between a range where the perturbative expansion is stable $C_k/\Lambda^2 \in [-4\pi, 4\pi]$ TeV$^{-2}$. The observed limit intervals are reported in Table 1 for the independent and marginalized scenarios. The expected limits are compatible with the observed ones. Interestingly, the intervals obtained in the two scenarios are highly similar, showing that little correlation is present between the operators considered.

**Table 1.** Observed intervals at 95% CL for the coupling parameters of the four independent EFT operators contributing to $t\bar{t}t\bar{t}$ production. Intervals are reported for both the independent and marginalized scenarios [28].

| Coupling Parameter | Marginalized $C_k/\Lambda^2$ (TeV$^{-2}$) | Independent (TeV$^{-2}$) |
|:---:|:---:|:---:|
| $C_{\mathcal{O}_{tt}^1}$ | $[-2.2, 2.1]$ | $[-2.1, 2.0]$ |
| $C_{\mathcal{O}_{QQ}^1}$ | $[-2.2, 2.0]$ | $[-2.2, 2.0]$ |
| $C_{\mathcal{O}_{Qt}^1}$ | $[-3.7, 3.5]$ | $[-3.5, 3.5]$ |
| $C_{\mathcal{O}_{Qt}^8}$ | $[-8.0, 6.8]$ | $[-7.9, 6.6]$ |

Emerging Machine Learning techniques to probe EFT operators [55], such as those used by CMS in [56], provide a very promising path towards interpreting $t\bar{t}t\bar{t}$ searches in the context of EFT theories and take into account the effect of the operators on the event kinematics.

*3.3. BSM Sensitivity*

The rate of $t\bar{t}t\bar{t}$ production may be significantly enhanced in several BSM models. For example, new particles that couple to the top quark and have masses two-fold greater than the top quark mass, such as heavy scalars or pseudoscalars predicted in Type-II two-Higgs-doublet models (2HDM) [57–59] or simplified models of dark matter [60,61], can be produced on-shell in association with top quarks and subsequently decay into top quark pairs. This results in an increased $t\bar{t}t\bar{t}$ production cross section. Less-massive particles, such as a scalar ($\phi$) or vector boson ($Z'$) with couplings to the top quark [62], may also enhance the $t\bar{t}t\bar{t}$ cross section through off-shell contributions. Final states with four top quarks may also be produced through the decay of pair-produced gluinos in models of supersymmetry [63–72]; however, for sufficiently massive gluinos (>1 TeV), these are typically studied in searches requiring very large missing transverse momentum and boosted signatures [73–76].

The CMS search for $t\bar{t}t\bar{t}$ production in same-charge dilepton and multi-lepton final states [26] described in Section 2.1 has reported limits on the masses and couplings of a neutral $\phi$ or $Z'$ with masses smaller than $m_t$ that could contribute to $\sigma(t\bar{t}t\bar{t})$ through off-shell effects. Couplings larger than 1.2 are excluded for $m_\phi$ between 25 and 340 GeV. For $m_{Z'} = 25$ (300) GeV, couplings larger than 0.1 (0.9) are excluded. The search also probes models with new scalar or pseudoscalar (H/A) particles with masses greater than $m_t$ decaying to $t\bar{t}$ and produced in association with a single-top quark or a top quark pair. Limits are placed in the plane of $\tan\beta$ vs. $m_{H/A}$ for Type-II 2HDM models in the alignment limit [77,78]. For $\tan\beta = 1$, H (A) masses up to 470 (550) GeV are excluded. Similar exclusions are placed on simplified models of DM with a Dirac fermion DM candidate ($\chi$) in addition to H/A when setting the parameters $g_{\rm SM}$ and $g_{\rm DM}$, representing the couplings of H/A to SM fermions and $\chi$, respectively, to 1, and assuming $m_{H/A} < m_\chi$. Large portions of the parameter space of $m_\chi$ vs. $m_{H/A}$ are excluded when relaxing the $m_{H/A} < m_\chi$ assumption for specific choices of $g_{\rm DM} = 1$ or 0.5 with $g_{\rm SM} = 1$.

Four top quark production is also relevant in probing the existence of color-octet scalar states, commonly referred to as sgluons, which are predicted in some models of new physics such as non-minimal supersymmetric models featuring Dirac gauginos. In the supersymmetric case, a complex color-octet scalar is predicted that splits into two non-degenerate real components after SUSY breaking, a scalar and a pseudoscalar in the case that its couplings preserve CP [79]. The pseudoscalar, generally expected to be lighter, decays solely into quark pairs and predominantly into $t\bar{t}$, while the scalar, generally heavier, decays into both quarks and gluons. Sgluon production and decay could thus contribute to $t\bar{t}t\bar{t}$ production. CMS results probing $t\bar{t}t\bar{t}$ production in the same-charge dilepton and multi-lepton final states with 35.9 fb$^{-1}$ of data collected at $\sqrt{s} = 13$ TeV [80] have been used to place constraints on sgluon pair production, conservatively excluding pseudoscalar sgluon masses up to 1.06 TeV at 95% CL [79]. The sensitivity to sgluon production can be improved in future measurements by adopting a dedicated search strategy exploiting the kinematic properties of the signal, such as features in the distribution of hadronic activity [79].

## 4. Future of Four-Top Quark Measurements

The high-luminosity LHC is expected to provide a fertile environment for $t\bar{t}t\bar{t}$ studies [23]. While the production cross section increases by a modest factor of 1.3 when the centre of mass energy of pp collisions is increased from 13 to 14 TeV (and by a factor of 1.19 from 13 TeV to 13.6 TeV), the signal-to-background ratio is expected to improve since this increase is smaller for most backgrounds. Four-top quark production also shows promise at the higher-energy future colliders currently under study, such as the HE-LHC ($\sqrt{s} = 27$ TeV) and FCC-hh ($\sqrt{s} = 100$ TeV) [25]. Moreover, the high-collision energies also have the consequence that partons at lower Björken $x$ values will be in the phase space for $t\bar{t}t\bar{t}$ production. This means that the theoretical uncertainties originating from sources such as parton density functions are expected to become substantially reduced, even after considering the lack of improvements beyond the current state of the art. At the HL-LHC, HE-LHC and FCC-hh, $t\bar{t}t\bar{t}$ measurements bear the potential for precision QCD tests and precise physics measurements including stringent SMEFT constraints on four-quark interactions [81].

With 3 ab$^{-1}$ of integrated luminosity collected at the HL-LHC, analyses using leptonic final states will start relying on detailed prediction of the SM backgrounds that create same-charge leptons or multi-lepton backgrounds, such as $t\bar{t}V$ and multi-boson production. ATLAS projects that the $t\bar{t}t\bar{t}$ production cross section can be constrained to 11% total accuracy using events with two same-charge leptons or at least three leptons [24,82]. In the same final state and with the same luminosity, CMS expects the statistical uncertainty of a cut-and-count analysis to be of the order of 9% but warns that backgrounds estimated from simulation introduce substantial systematic uncertainties between 18% and 28% depending on the considered sources of theory uncertainty. At the HE-LHC, a similar analysis could be expected to constrain the $t\bar{t}t\bar{t}$ production cross section to within a 1–2% statistical uncertainty, and the systematic uncertainties also decrease due to the improved signal to background ratio [24,83]. A more recent ATLAS extrapolation [84] based on the Run 2 result described in Section 2.1 with different scenarios for the improvement of the systematic uncertainties, projects a $t\bar{t}t\bar{t}$ cross section uncertainty of 14% for the most optimistic case at the HL-LHC.

When the $t\bar{t}t\bar{t}$ production cross section is constrained to this accuracy, the measurements can again be employed to constrain the top-Higgs interaction. Using the same-charge and multi-lepton cross section values, the modification factor that quantifies the Higgs contribution to $\sigma_{t\bar{t}t\bar{t}}$, $\kappa_t$, can be estimated using the projected cross-section uncertainties. Assuming that the cross section is modified but acceptance and analysis efficiency do not change substantially, a direct bound on $\kappa_t \leq 1.41$ can be obtained at the HL-LHC and $\kappa_t \leq 1.15(1.12, 1.10)$ with a luminosity of 10 (20, 30) ab$^{-1}$ at the HE-LHC, respectively. The measurement of $\kappa_t$ provides a direct link to the top quark Yukawa coupling; however, it should be noted that these estimates are dependent on the order of theoretical calculations

and were not determined using the complete NLO calculations [24,44]. A similar procedure can also be performed for modifications to $\sigma_{t\bar{t}t\bar{t}}$ from SMEFT contributions [24,83]. Depending on the operator, constraints for top-up quark operators can be very tight at the HL-LHC, down to $|\tilde{C}_{tu}^{(1)}| < 2.5$, and for generic top-quark interactions down to $|\tilde{C}_{tq}^{(1)}| < 2.2$. The four-top quark interaction coefficients can be constrained even more tightly down to approximately $|\tilde{C}_{tt}| < 1.1$ at the HL-LHC or well below 1.0 for the HE-LHC, a substantial improvement compared to Table 1. The production of $t\bar{t}t\bar{t}$ can also be used to constrain the top quark dipole moment [85].

Many of the BSM theories that predict final states with $t\bar{t}t\bar{t}$ can be investigated at the HL-LHC. Most projections for these searches were performed in the high-purity leptonic final states and were limited by the current knowledge of $t\bar{t}V$ and $t\bar{t}t\bar{t}$ production, and could potentially be also explored in other final states for enhanced sensitivity. A study by ATLAS [86] investigates $t\bar{t}t\bar{t}$ at the HL-LHC in same-charge lepton and multi-lepton signatures to search for two additional scalars that can both decay to $t\bar{t}$ or enhance $t\bar{t}t\bar{t}$ production, and where $t\bar{t}t\bar{t}$ and $t\bar{t}V$ production would be the dominant background. These studies project that scalar dark matter mediators $A$ and $H$ from the previously mentioned two-Higgs doublet models can be observed with sensitivity for $A$ masses between a few 100 GeV and 1 TeV for $m_H$ = 600 GeV and $\sin\theta = 0.35$, or excluded over large range of $\sin\theta$ values for lower $m_H$. When extrapolating $t\bar{t}t\bar{t}$ production in a recast of a cut-and-count analysis by the CMS experiment in same-charge and multi-lepton final states, sgluons and similar coloured pseudoscalar octet particles could be excluded for masses under 1260 and 1470 GeV, respectively, for the HL-LHC and HE-LHC full datasets [24,79,87].

It is worth mentioning that a higher-energy hadron collider, such as the FCC-hh [25], would offer opportunities for an extremely diverse $t\bar{t}t\bar{t}$ measurement program. However, these studies are still very much in their infancy and are also beyond the scope of this review.

*Opportunities*

The production of three top quarks in the SM can occur in association with a light quark or a W boson [88,89]. One of the tree-level diagrams contributing to the $t\bar{t}tj$ process, where $j$ is a light quark, is mediated by the triple gauge boson vertex. Despite a less busy final state with respect to $t\bar{t}t\bar{t}$, the production of three top quarks in the SM at the LHC at a centre-of-mass energy of 13 TeV is far rarer with $\sigma_{t\bar{t}tW} = 0.73$ fb and $\sigma_{t\bar{t}tj} = 0.47$ fb. While this process, unless enhanced by BSM physics, is very likely outside of the reach of LHC Run 2 and the upcoming Run 3 data-collecting periods, the potential exists for finding evidence of the process with the HL-LHC and HE-LHC full data sets [89]. Production of three top quark without extra jets or W bosons requires flavour changing neutral currents. Setting limits on the production of this process can help to constrain *uttt* EFT operators according to [90,91].

Final states with one or more hadronically decaying $\tau$ leptons constitute $\approx$29% of $t\bar{t}t\bar{t}$ decays, and are not currently being exploited by the LHC searches. The exploration of these decay modes presents an interesting opportunity for future investigations, and could be relevant for interpretations in certain leptoquark models. Such models may be interesting in light of the anomalies observed in lepton flavour universality measurements.

## 5. Conclusions

In this paper, we reviewed the current status of searches for $t\bar{t}t\bar{t}$ production at hadron colliders. In particular, recent searches from the ATLAS and CMS Collaborations are summarised. The searches were performed using data collected over the time period spanning from 2016 to 2018 exploring several final states with one, two (same charge or opposite charge) and multiple leptons in the final state. Combinations of different final states, advanced machine learning techniques, and innovative background estimation techniques provide evidence of $t\bar{t}t\bar{t}$ production at 4.7 standard deviations in a measurement from ATLAS [22]. The most precise estimations of the cross section are $24 \pm 4(\text{stat})^{+5}_{-4}(\text{syst})$ fb

= $24^{+7}_{-6}$ fb and $\sigma(t\bar{t}t\bar{t}) = 17 \pm 5$ (stat + syst) fb by the ATLAS and CMS collaborations, respectively. These measurements are statistically consistent and can be compared with the current highest-order theoretical cross section of $12.0 \pm 2.4$ fb cross section at a centre-of-mass energy of $\sqrt{s} = 13$ TeV.

The $t\bar{t}t\bar{t}$ process can be exploited to measure relevant parameters of the SM and its effective field theory extension. The Higgs-mediated production diagram of $t\bar{t}t\bar{t}$ exposes the Yukawa coupling of the top quark, which is measured to be $|y_t/y_{SM}| < 1.7$ at 95% CL in a CMS analysis [26]. Effective field theory operators involving four heavy quarks or two heavy and two light quarks were constrained exploring the effect of such operators on the $t\bar{t}t\bar{t}$ cross section.

Additionally, the $t\bar{t}t\bar{t}$ process offers a direct portal to physics beyond the standard model. Models introducing additional light neutral scalar ($\phi$) and vector ($Z'$), or heavy ($m > 2m_t$) scalar ($H$) and pseudoscalar ($A$) bosons in the context of 2HDM models have been constrained in a CMS analysis [26]. In the context of SUSY models, constraints can be placed on sgluon pair production.

The future of the LHC program and its high-luminosity and high-energy upgrades provide opportunities for precise SM and EFT measurements and searches for new physics with the $t\bar{t}t\bar{t}$ process. Projections of current results for the HL- and HE-LHC programs predict the ability to measure the $t\bar{t}t\bar{t}$ cross section with a precision of 11% using the full data set of the HL-LHC and constrain $|y_t/y_{SM}| < 1.1$ at the HE-LHC. The exploration of $t\bar{t}t\bar{t}$ final states with $\tau$ leptons or no leptons (all-hadronic) will further enrich the four top quark physics program. Finally, the large integrated luminosity accumulated by the LHC project and its extension will allow researchers to explore even rarer related processes, for which $t\bar{t}t\bar{t}$ is a background process such as $t\bar{t}tV$ and $t\bar{t}tq$ production.

Overall, the $t\bar{t}t\bar{t}$ process offers a wide breadth of opportunities for building a strong physics program including both precise measurements of important SM parameters and its EFT extensions, and the direct probing of different types of BSM theories. Although the $t\bar{t}t\bar{t}$ research program has commenced only recently, highly promising results have already been obtained with the current data collected by the LHC.

**Funding:** This research received no external funding.

**Acknowledgments:** We kindly thank the editors of this dedicated issue of Universe for the invitation to contribute this overview article. F.B. acknowledges support from DESY (Hamburg, Germany) a member of the Helmholtz Association HGF, and support by the Deutsche Forschungsgemeinschaft (DFG, German Research Foundation) under Germany's Excellence Strategy—EXC 2121 "Quantum Universe"—390833306. V.D. acknowledges support from the US Department of Energy Grant Number DE-SC0011702 to University of California Santa Barbara, Santa Barbara, CA, USA. E.U. acknowledges support from the US Department of Energy Grant Number DE-SC0010010 to Brown University, Providence, RI, USA. F.D. acknowledges the support of CEA-DRF/IRFU, France.

**Conflicts of Interest:** The authors declare no conflict of interest.

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
