# Peer review of "Four-top quark physics at the LHC"

_universe, doi:10.3390/universe8120638_

Round 1
Reviewer 1 Report
The manuscript by Blekman et al provides a nice review and summary of the experimental status of four-top quark production at the LHC. The article is well written and accessible. I recommend its publication in Universe. I only have a few suggestions to improve the paper:
1. The authors could include a figure with LO Feynman diagrams for the process, and perhaps say a few more things about the theoretical calculations.
2. References are given to the discovery of the top quark in 1995 but it would also be nice to give references to the first observation of single-top production at the Tevatron (not just the later measurements at the LHC).
3. In the references would it not be more appropriate to also write the name of the collaboration, instead of just the first author followed by "others"?
Author Response
Comments and Suggestions for Authors The manuscript by Blekman et al
provides a nice review and summary of the experimental status of four-top
quark production at the LHC. The article is well written and accessible. I
recommend its publication in Universe.
The authors thank the referee for their positive review and constructive
comments. We hope the referee agrees that including a very recent CMS result
in this review was worth the slight delay. Please see below for detailed replies.
I only have a few suggestions to improve the paper:
1. The authors could include a figure with LO Feynman diagrams for the
process, and perhaps say a few more things about the theoretical calculations.
Authors answer: We agree, and have included two representative diagrams
out of the over 100 leading order diagrams available. We believe that it
would be very difficult to correctly represent the challenges of these theoretical
calculations within the scope of this experimentally-oriented review.
2. References are given to the discovery of the top quark in 1995 but
it would also be nice to give references to the first observation of single-top
production at the Tevatron (not just the later measurements at the LHC).
Authors Answer: We agree. And have included the references
3. In the references would it not be more appropriate to also write the
name of the collaboration, instead of just the first author followed by ”oth-
ers”?
Authors Answer: We agree. To achieve this we however had to over-
write the default bibliography configuration file of MDPI. We hope they don’t
mind, anyway having a configuration that is prepared with dealing with large
collaboration author list papers is a good thing. This may also be relevant
for unpublished documents such as ATLAS Conf notes and CMS Physics
analysis summaries.
Reviewer 2 Report
Dear Editor,
The chapter of this review explains very clearly the state of the art of experimental analyses in the context of four-top quark physics at the LHC. It also discusses interpretations and prospects for future hadron colliders. The text is comprehensive and well written.
I would definitely accept this chapter for publication, but there are a few minor suggestions I would like to give. I am going to list them in the following:
- Citations of theoretical works can be improved.
Regarding four top, when [13-15] is cited at line 24 also
Maltoni:2015ena, Frederix:2018nkq, Jezo:2021smh should be cited.
Regarding ttV ew effects, together with [24] at line 88 also
Frederix:2017wme (the ref 13), Frederix:2018nkq, Broggio:2019ewu, Kulesza:2020nfh, vonBuddenbrock:2020ter, FebresCordero:2021kcc, Bevilacqua:2021tzp, Denner:2021hqi should be cited
- I believe the reader would appreciate an explanation or at least a motivation of why the required minimum number of light jets is so different in ATLAS and CMS analyses (lines 74-75 and 163-166).
- It would be easier for the reader if the cuts for the event selections were summarised in formulas or a tables. This applies for both sections 2.1.1 and 2.2.1 and separately the ATLAS and CMS cases.
- In section 2.1.2, for CMS, part of the variables entering the BDT have been specified, but the total number of them is unknown (line 142).
Instead, for ATLAS, it is said that they are 12 (line 126), but none of them is specified. The description could be harmonised using the same style for ATLAS and CMS and listing the variables, maybe mentioning those that give the highest sensitivity could be also useful.
- In section 3.1 a comparison with other methods for the extraction of yt (ttH and perhaps virtual effects in tt) should be discussed. At least the results obtained with the other methods should be mentioned and compared to the four-top case.
- In section 3.2 and in general when EFT operators are mentioned, rather than giving just the conventional names of the operators, it would be better defining the operators so that any reader would know what they are.
- At line 282-283 it is said: “The value is competitive with on-shell Higgs boson analyses”. It would be better to report the actual values together with the citations to the corresponding analyses.
Best regards,
the referee.
Author Response
1.2 Referee 2:
Dear Editor,
The chapter of this review explains very clearly the state of the art of
experimental analyses in the context of four-top quark physics at the LHC.
It also discusses interpretations and prospects for future hadron colliders.
The text is comprehensive and well written.
I would definitely accept this chapter for publication, but there are a few
minor suggestions I would like to give.
The authors thank the referee for their positive review and constructive
comments, particularly the bibliography recommendations are wholeheartedly
received. We hope the referee agrees that including a very recent CMS result
in this review was worth the slight delay. Please see below for detailed replies.
I am going to list them in the following:
- Citations of theoretical works can be improved. Regarding four top,
when [13-15] is cited at line 24 also Maltoni:2015ena, Frederix:2018nkq,
Jezo:2021smh should be cited.
Regarding ttV ew effects, together with [24] at line 88 also Frederix:2017wme
(the ref 13), Frederix:2018nkq, Broggio:2019ewu, Kulesza:2020nfh, vonBud-
denbrock:2020ter, FebresCordero:2021kcc, Bevilacqua:2021tzp, Denner:2021hqi
should be cited
Authors answer: These are all incorporated. Thank you.
- I believe the reader would appreciate an explanation or at least a mo-
tivation of why the required minimum number of light jets is so different in
ATLAS and CMS analyses (lines 74-75 and 163-166).
Authors answer: this is driven by the fact that the two collaborations have
a very different strategy regarding background-dominated regions. These are included in the ATLAS analysis but used as control regions where corrections
are derived for the final signal-region, and included in the CMS analyses
and used to constrain systematic uncertainties through nuisance parameters.
We have included a short sentence in each of the locations to elucidate this
different philosophy.
- It would be easier for the reader if the cuts for the event selections were
summarised in formulas or a tables. This applies for both sections 2.1.1 and
2.2.1 and separately the ATLAS and CMS cases.
Authors answer: As the selection cuts are so dependent of the analysis
strategy, the authors would like to decline. Without the complex multivariate
techniques and systematic uncertainty treatment, the exact values of the event selection cuts are effectively not relevant as the overwhelming majority of the selected events are background, not t ̄tt ̄t signal.
- In section 2.1.2, for CMS, part of the variables entering the BDT have
been specified, but the total number of them is unknown (line 142). Instead,
for ATLAS, it is said that they are 12 (line 126), but none of them is specified.
The description could be harmonised using the same style for ATLAS and
CMS and listing the variables, maybe mentioning those that give the highest
sensitivity could be also useful.
Authors answer: We have added some information to unify the descrip-
tion between the experiments. The number of input variables for deep learn-
ing MVAs (so also BDTs) is not so important as long as variables are not
100% (anti)correlated or introduce noise due to not having any discriminat-
ing power. Both ATLAS and CMS use relatively standard information such
as b-tagging and top-tagging information, event kinematic such as lepton/jet
momenta and angles, and event level variables such as HT and √s. In gen-
eral HT and information related to b- and top-tags are the most sensitive,
but we consider this to much information for this overview, particularly as
the number of input variables can differ drastically per year/dataset even for
MVAs by the same collaboration.
- In section 3.1 a comparison with other methods for the extraction of
yt (ttH and perhaps virtual effects in tt) should be discussed. At least the
results obtained with the other methods should be mentioned and compared
to the four-top case.
Authors answer: we have added more details on yukawa coupling mea-
surements and cited the most recent results on yt. The value is comparable
but slightly weaker than compared to other indirect measurements (such as
from ttbar+jets production), while the SM ttH measurement of yt makes as-
sumptions about the exchanged boson which these indirect measurements from
tttt and tt+jets do not include.) - In section 3.2 and in general when EFT operators are mentioned, rather
than giving just the conventional names of the operators, it would be better
defining the operators so that any reader would know what they are. Authors answer: We have expanded the introduction of this section to provide a more didactic explanation of the naming convention. - At line 282-283 it is said: “The value is competitive with on-shell Higgs
boson analyses”. It would be better to report the actual values together with
the citations to the corresponding analyses.
Authors answer: We now include values in the text.
Reviewer 3 Report
This article has very intense information and summarized work of ATLAS and CMS experiments about about four top quark physics.
I accept to publish this article as a review paper.
Author Response
This article has very intense information and summarized work of ATLASand CMS experiments about about four top quark physics.
I accept to publish this article as a review paper.
The authors thank the referee for their positive review. We hope the
referee agrees that including a very recent CMS result in this review was
worth the slight delay.